# FGFR Signaling as a Candidate Therapeutic Target for Cancers Resistant to Carbon Ion Radiotherapy

**DOI:** 10.3390/ijms20184563

**Published:** 2019-09-14

**Authors:** Narisa Dewi Maulany Darwis, Ankita Nachankar, Yasushi Sasaki, Toshiaki Matsui, Shin-ei Noda, Kazutoshi Murata, Tomoaki Tamaki, Ken Ando, Noriyuki Okonogi, Shintaro Shiba, Daisuke Irie, Takuya Kaminuma, Takuya Kumazawa, Mai Anakura, Souichi Yamashita, Takashi Hirakawa, Sangeeta Kakoti, Yuka Hirota, Takashi Tokino, Akira Iwase, Tatsuya Ohno, Atsushi Shibata, Takahiro Oike, Takashi Nakano

**Affiliations:** 1Department of Radiation Oncology, Gunma University Graduate School of Medicine, Maebashi 371-8511, Japan; narisajimmy@gmail.com (N.D.M.D.); drankitan@gmail.com (A.N.); matsu.toshi42@gmail.com (T.M.); nodashin@saitama-med.ac.jp (S.-e.N.); kazutoshi.m@gmail.com (K.M.); tamakit@fmu.ac.jp (T.T.); ken.ando0906@gmail.com (K.A.); okonogi.noriyuki@qst.go.jp (N.O.); shiba4885@yahoo.co.jp (S.S.); daisuke.i0626@gmail.com (D.I.); cami_taku@yahoo.co.jp (T.K.); takuya.kumazawa@gmail.com (T.K.); mai.anakura@gmail.com (M.A.); drsangeeta84@gmail.com (S.K.); yukahirota@gunma-u.ac.jp (Y.H.); tnakano@gunma-u.ac.jp (T.N.); 2Department of Medical Genome Sciences, Research Institute for Frontier Medicine, Sapporo Medical University, Sapporo 060-8556, Japan; yasushi@sapmed.ac.jp (Y.S.); tokino@sapmed.ac.jp (T.T.); 3Department of Obstetrics and Gynecology, Gunma University Graduate School of Medicine, Maebashi 371-8511, Japan; soichi-y@gunma-cc.jp (S.Y.); tahirakawa@gmail.com (T.H.); akiwase@gunma-u.ac.jp (A.I.); 4Gunma University Heavy Ion Medical Center, Maebashi 371-8511, Japan; tohno@gunma-u.ac.jp; 5Gunma University Initiative for Advanced Research (GIAR), Maebashi 371-8511, Japan; shibata.at@gunma-u.ac.jp

**Keywords:** carbon ion radiotherapy, uterine cervical cancer, next-generation sequencer, somatic mutations, *FGFR*, radiosensitization, LY2874455

## Abstract

Radiotherapy is an essential component of cancer therapy. Carbon ion radiotherapy (CIRT) promises to improve outcomes compared with standard of care in many cancers. Nevertheless, clinicians often observe in-field recurrence after CIRT. This indicates the presence of a subset of cancers that harbor intrinsic resistance to CIRT. Thus, the development of methods to identify and sensitize CIRT-resistant cancers is needed. To address this issue, we analyzed a unique donor-matched pair of clinical specimens: a treatment-naïve tumor, and the tumor that recurred locally after CIRT in the same patient. Exon sequencing of 409 cancer-related genes identified enrichment of somatic mutations in *FGFR3* and *FGFR4* in the recurrent tumor compared with the treatment-naïve tumor, indicating a pivotal role for FGFR signaling in cancer cell survival through CIRT. Inhibition of FGFR using the clinically available pan-FGFR inhibitor LY2874455 sensitized multiple cancer cell lines to carbon ions at 3 Gy (RBE: relative biological effectiveness), the daily dose prescribed to the patient. The sensitizer enhancement ratio was 1.66 ± 0.17, 1.27 ± 0.09, and 1.20 ± 0.18 in A549, H1299, and H1703 cells, respectively. Our data indicate the potential usefulness of the analytical pipeline employed in this pilot study to identify targetable mutations associated with resistance to CIRT, and of LY21874455 as a sensitizer for CIRT-resistant cancers. The results warrant validation in larger cohorts.

## 1. Introduction

Radiotherapy is an essential component of cancer therapy [1]. Carbon ion radiotherapy (CIRT) is an advanced radiotherapy modality that holds great promise [2]. CIRT has two advantages over conventional photon radiotherapy: sharper dose distribution, and higher cell-killing ability [2]. Evidence shows better treatment outcomes for CIRT compared with standard of care for many cancers, including those known to be resistant to conventional photon radiotherapy [3]. Nevertheless, patients often experience in-field recurrence after CIRT, indicating the presence of a subset of cancers that harbor intrinsic resistance to CIRT. Therapeutic regimens for CIRT are currently undergoing optimization, where not a small proportion of cases is treated by monomodality. Methods are needed to identify and sensitize CIRT-resistant cancers. However, targetable biological properties in cancers associated with resistance to CIRT have not been elucidated fully. To address this issue, we conducted a pilot study to investigate the mutation profiles of unique clinical cancer specimens: a pair of treatment-naïve tumors, and the tumor that recurred locally after CIRT in the same patient. By performing an exon sequencing of cancer-related genes, we identified enrichment of the FGFR signaling pathway in the CIRT-recurrent tumor. We further showed the sensitizing effect of the clinically available pan-FGFR inhibitor LY2874455 on carbon ions in multiple cancer cell lines.

## 2. Results

To explore targetable biological properties in CIRT-resistant cancers, we analyzed the genetic profile of tumors collected from a patient with uterine cervical cancer who experienced local recurrence after CIRT (Table 1). The patient was 45 years old and diagnosed with squamous cell carcinoma of the uterine cervix, cT2bN1M0 according to the UICC TNM classification of malignant tumors, 6th edition. The tumor was positive for human papillomavirus 16. The patient received CIRT at the Gunma University Heavy Ion Medical Center in a prospective phase I trial for CIRT in uterine cervical cancer (UMIN000013340). The details of the trial, including eligibility and treatment contents, are described in a previous report [4]. Tumors were biopsied at the time of diagnosis (T1) and of recurrence after CIRT (T2) (Table 1, Appendix A, and Section 4.5). Exons of 409 genes related to cancers were sequenced using the Ion AmpliSeq Comprehensive Cancer Panel (CCP: Thermo Fisher Scientific, Waltham, MA, USA). The results of quality checking are summarized in Appendix A. In total, 87 somatic nonsynonymous mutations were identified.

Mutations in *PIK3CA* (E545K) were identified in both T1 and T2. These mutations were validated by Sanger sequencing (Figure 1A). *PIK3CA* E545K is one of the most prevalent somatic mutations in uterine cervical cancer [5,6,7]. In addition, the patterns of single-nucleotide substitutions in a three-base context were consistent between T1 and T2 (Figure 1B) [8]. This is in line with a previous study which demonstrated that the mutation spectra of single-nucleotide variations are fairly consistent across tumors that arise in the same organ of the same individuals [9]. Together, these data suggest the reliability of the sequencing data obtained with our analytical workflow.

Previous studies have shown that solid tumors are genetically heterogeneous, and that the small populations of cells resistant to treatment might already exist before initiation of treatment; the treatment acts as selection pressure, allowing the intrinsically treatment-resistant subpopulation to grow throughout the treatment [10,11]. Based on this concept, we analyzed the enrichment of somatic mutations through CIRT by comparing variant frequencies (VF). Subtraction of VF_T1_ from VF_T2_ showed a positive value for 13 genes including *SEDT2*, *CDH2*, *NUMA1*, *MTRR*, *FGFR3*, *FGFR4*, and *MYH11* (Figure 2A). The data indicate that the subpopulations harboring mutations in these genes were selectively enriched post-CIRT, therefore these mutations may be associated with resistance to CIRT. In addition, gene ontology (GO) analysis showed that the enriched genes were highly associated with the GO annotation *response to stimulus* (Figure 2B). This is reasonable, considering the fact that T2 survived in response to carbon ion irradiation. The enriched genes were also highly associated with the GO annotations *cell differentiation*, *protein metabolism*, and *signaling*, while they were less associated with other GO annotations, including *immune systems*, *cell proliferation*, and *cell death*.

Among these results, *FGFR3* and *FGFR4* caught our attention because the two genes were involved in the same pathway and encoded receptor tyrosine kinases, a targetable class of proteins [12], and the magnitude of mutation enrichment was among the highest (i.e., 40%, Figure 2A) when combined. Based on these data, we hypothesized that FGFR signaling is a possible target for sensitization of CIRT-resistant cancers. To test this, we evaluated the sensitizing effect of LY2874455 on carbon ions in CIRT-resistant cancer cells in vitro. LY2874455 is a pan-FGFR inhibitor available clinically [13,14]. As a model of CIRT-resistant cancer cells, we chose the A549, H1299, and H1703 cell lines based on previous studies that screened in vitro sensitivity to carbon ions in a panel of human cancer cell lines, demonstrating high carbon ion resistance for these cell lines [15,16]. LY2874455 suppressed phosphorylation of extracellular signal-regulated kinase 1/2 (ERK), a major downstream molecule in FGFR signaling [13], in a concentration-dependent manner (Figure 3A,B). In accordance with this, treatment with LY2874455 alone decreased clonogenic survival of non-irradiated cells in a concentration-dependent manner (Figure 3C). Based on these data, we chose 40 nM, a concentration yielding mild cytotoxicity, for radiosensitizing experiments. For the carbon ion dose, we chose 3 Gy (RBE: relative biological effectiveness) because 3 Gy (RBE) was prescribed daily for pelvic irradiation for the patient analyzed in this study. LY2874455 significantly enhanced carbon ion induced clonogenic cell death in all cell lines tested (Figure 4 and Table 2). The sensitizer enhancement ratio (SER) was 1.66 ± 0.17, 1.27 ± 0.09, and 1.20 ± 0.18 for A549, H1299, and H1703, respectively (Table 2). Together, these data indicate that FGFR signaling contributes to cancer cell resistance to CIRT and is targetable using LY2874455.

## 3. Discussion

This is the first study to report somatic mutations in tumors that recurred locally after CIRT. The results show enrichment of FGFR signaling post-CIRT, indicating a pivotal role for this pathway in cancer cell survival through CIRT. We previously reported activating mutations of *KRAS*, as well as amplification of *FGFR2*, in a tumor that exhibited extreme resistance to photon radiotherapy [17]. Other studies of hematological malignancies showed enrichment of the RAS/MAPK pathway in cancers that recurred after initial treatment with tyrosine kinase inhibitors [18,19]. Furthermore, an in vitro microarray analysis that compared radioresistant cells with radiosensitive cells demonstrated upregulation of *FGFR3* in the former [20]. The RAS/MAPK pathway is one of the major downstream pathways of FGFR, and ERK is a signal transduction molecule in the MAPK pathway [21]. These findings together indicate that upregulation of the FGFR–RAS–MAPK axis plays a pivotal role in cancer cells that survive potent cytotoxic treatment.

This study is also the first to report the radiosensitizing effect of LY2874455. LY2874455 is an orally administered pan-FGFR inhibitor that suppresses FGFR signaling, including *FGF2*- and *FGF9*-induced phosphorylation of ERK, by occupying the ATP-binding pocket in the kinase domains [13]. LY2874455 inhibits the proliferation of multiple cancer cell lines and tumor xenografts representing the major relevant FGF/FGFR histologies, including nonsmall cell lung cancers, gastric cancers, bladder cancers, and multiple myeloma [13]. A phase 1 study demonstrated fair tolerability and promising antitumor activity in patients with solid tumors [14]. In our study, LY2874455 sensitized carbon ions at a concentration that yielded a mild inhibitory effect on ERK phosphorylation, leading to modest cytotoxicity by itself. These data indicate the potential usefulness of LY2874455 as a CIRT sensitizer. This should be validated in xenograft models.

Normal tissue tolerance is an important issue with carbon ion radiotherapy. From this perspective, it should be noted that the normal tissue toxicity for this patient was tolerable, as reported previously [4].

The following issues are raised as limitations of this study. Firstly, the radiosensitization experiments shown in Figure 4 should ideally have been conducted using patient biopsy specimens. In the absence of such data, this study would be theoretical. However, conducting radiosensitization experiments using patient biopsy specimens is practically difficult. One major reason for this is that the plating efficiency of ex vivo cancer cells is very low (i.e., <10% in general), even in the absence of irradiation or drug treatment, making it difficult to obtain reliable radiosensitivity data. Future efforts are warranted for this issue. Secondly, the panel used in this study did not cover all genes previously reported to be mutated in uterine cervical cancers [22]. Thus, whole exome sequencing should be performed in future to obtain a greater understanding of CIRT resistance in uterine cervical cancer. Thirdly, we analyzed only one case due to the rarity of patients who receive CIRT at a site accessible for biopsy of recurrent tumors. Lastly, we did not investigate the mechanisms underlying the sensitizing effect of LY2874455 because that was out of our research scope.

In summary, we showed enrichment of mutations in genes involved in FGFR signaling in a CIRT-resistant tumor, as well as sensitization of carbon ions by the clinical pan-FGFR inhibitor LY2874455. The analytical pipeline employed in this pilot study was useful for the identification of targetable mutations associated with resistance to CIRT, and this approach warrants validation in larger cohorts.

## 4. Materials and Methods

### 4.1. Ethics

This study was approved by the Institutional Review Boards of Gunma University Hospital (approval number: 1109) on 27 November, 2013. Written informed consent was obtained from the patient. The study was conducted in accordance with the ethical principles of the Declaration of Helsinki.

### 4.2. Tissue Sample Collection

Tissue sample collection was performed as previously described [17]. The presence and the absence of malignant cells in the tumor and normal tissue samples, respectively, was confirmed histologically.

### 4.3. DNA Preparation

DNA preparation was performed as previously described [17]. In brief, DNA was extracted from formalin-fixed and paraffin-embedded (FFPE) tissues using the QIAamp DNA FFPE Tissue kit (Qiagen, Venlo, Netherlands). The TaqMan RNase P Detection Reagent (Thermo Fisher Scientific) was used to quantify purified DNA.

### 4.4. Next-Generation Sequencing

Sequencing was performed as previously described [17]. Forty nanograms of the tissue-extracted DNA was amplified by a multiplex polymerase chain reaction using the CCP, covering >95% of the exons of the 409 genes listed in Appendix A. After preparation for the library, sequencing was performed using an Ion Torrent next-generation sequencer (Thermo) using an Ion PI HI-Q Chef kit (Thermo) [23,24,25].

### 4.5. Identification of Somatic Mutations

Somatic mutations were determined as previously described [17]. In brief, the following criteria were used as cutoffs: total coverage >30, variant coverage >10, VF >5%, and minor allele frequency <0.1%. Possible strand-specific errors were filtered using the Integrative Genomics Viewer [26]. Single-nucleotide polymorphisms were excluded using the dbSNP database.

### 4.6. GO Enrichment Analysis

Using the Alliance of Genome Resource (https://www.alliancegenome.org), GO ribbons for the genes of interest were obtained. The color intensity for the annotations shown in the GO ribbons was interpreted as scores with a range of 0–4. The GO enrichment score was calculated by multiplying the color intensity score with the subtraction of T1 variant frequency from T2.

### 4.7. Cell Culture and Materials

Cell culture was performed as previously described [15,16]. In brief, A549, H1299, and H1703 were purchased from ATCC and cultured in RPMI-1640 (Sigma-Aldrich, St. Louis, MO, USA) supplemented with 10% fetal bovine serum (Life Technologies, Carlsbad, CA, USA). LY2874455 was purchased from Cayman Chemicals (Ann Arbor, MI, USA). LY2874455 stock solution was prepared by dissolving the original reagent in dimethyl sulfoxide at 20 mM and stored at −20°C. A working solution of LY2874455 was prepared before every experiment.

### 4.8. Carbon Ion Irradiation

Carbon ion irradiation was performed at the Gunma University Heavy Ion Medical Center as previously described [15,16], using the following beam specifications: 290 MeV/nucleon; average linear energy transfer at the center of a 6 cm spread-out Bragg peak of approximately 50 keV/μm; and vertical beam direction.

### 4.9. Clonogenic Assays

Clonogenic assays were performed as previously described [27]. In brief, the cells were seeded in 6-well plates. After incubation for 12 h, the medium was changed to a fresh medium containing LY28734455. After incubation for an additional 1 h, the cells were exposed to carbon ions. After incubation for 10 days, the cells were fixed with methanol and stained with crystal violet. Colonies comprising at least 50 cells were counted. The experiments were performed in quadruplicate. Surviving fractions were calculated after normalizing to unirradiated controls. SER was calculated by dividing the surviving fraction obtained in the presence of LY2874455 by that obtained in the absence of the drug.

### 4.10. Immunoblotting

Immunoblotting was performed as previously described [28]. The following antibodies were purchased from Cell Signaling Technology (Danvers, MA, USA): ERK (9107), phospho-ERK-Thr202/Tyr204 (4370), and GAPDH (3683). Uncropped versions of the immunoblots are shown in Appendix A. The immunoblot bands were quantitated using ImageJ (National Institutes of Health, Maryland, USA).

### 4.11. Statistical Analysis

Differences in clonogenic survival between two groups were assessed as follows. First, normality was confirmed by the Shapiro–Wilk test. For the data that followed normal distribution, variance was assessed by an F-test. Differences between two groups with equal variance were assessed by the Student’s t-test, while those without equal variance were assessed by Welch’s t-test. For the data that did not follow normal distribution, differences between two groups were assessed by the Mann−Whitney U-test. All statistical analyses were performed using Prism 8 (GraphPad Software, San Diego, CA). A *P*-value <0.05 was considered significant.

## Figures and Tables

**Figure 1 ijms-20-04563-f001:**
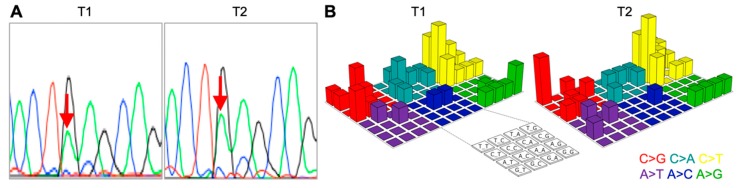
Quality assurance of next-generation sequencing data. (**A**) Verification of mutations in *PIK3CA* (c.1633G>A) by Sanger sequencing. Red arrows show variant. (**B**) Lego plots showing mutational patterns in a three-base context. The identified somatic single-nucleotide variants are grouped based on base substitution pattern and the neighboring bases.

**Figure 2 ijms-20-04563-f002:**
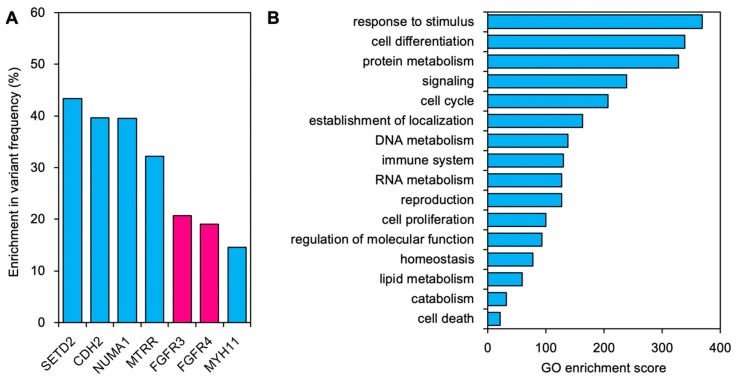
Enrichment analysis for somatic non-synonymous mutations comparing T2 with T1. (**A**) Subtraction of VF_T1_ from VF_T2_. The genes showing subtraction values >10% are listed. (**B**) Sum of gene ontology (GO) enrichment scores for the genes listed in A (see Section 4.6 for the calculation of GO enrichment scores).

**Figure 3 ijms-20-04563-f003:**
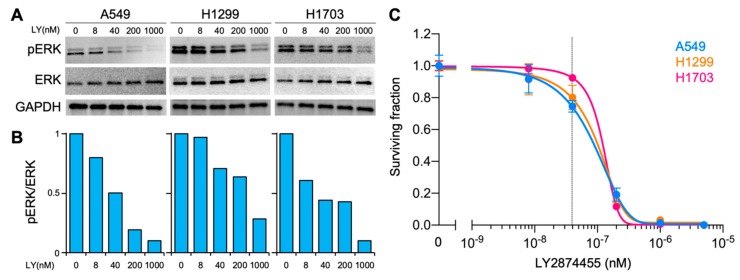
Concentration-dependent effect of LY2874455 (LY). (**A**) Immunoblots showing suppression of phosphorylation of extracellular signal-regulated kinase (ERK). Cells were exposed to LY for 1 h before collection. pERK, phosphorylated ERK. (**B**) Quantitation of immunoblots shown in A. The ratio of pERK to total ERK is shown relative to untreated controls and normalized to GAPDH. (**C**) Clonogenic survival of cells treated by LY28744554 alone. Dashed line indicates 40 nM.

**Figure 4 ijms-20-04563-f004:**
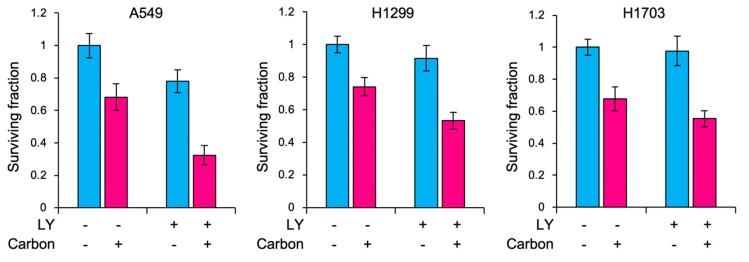
Sensitizing effect of LY2874455 on carbon ions as assessed by clonogenic assays. Cells were exposed to LY2874455 (LY, 40 nM) for 1 h and irradiated with carbon ions for 3 Gy (RBE).

**Table 1 ijms-20-04563-t001:** Clinical course of the patient and timing of sample collection.

Event	Treatment	Months	Sample
Diagnosis		0	T1
	CIRT	1	
Local recurrence		13	T2
	Surgery	15	Normal
Deceased		25	

CIRT: carbon ion radiotherapy.

**Table 2 ijms-20-04563-t002:** Sensitizing effect of LY2874455 on carbon ions.

Cell line	LY	SF at 3Gy (RBE)	*p*-Values	SER
A549	-	0.68 ± 0.06	0.00050	1.66 ± 0.17
	+	0.41 ± 0.05		
H1299	-	0.74 ± 0.06	0.00049	1.27 ± 0.09
	+	0.58 ± 0.03		
H1703	-	0.67 ± 0.05	0.026	1.20 ± 0.18
	+	0.56 ± 0.04		

LY: LY2874455 (40 nM); SF: surviving fraction; SER: sensitizer enhancement ratio.

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
