# Peer review of "FGFR Signaling as a Candidate Therapeutic Target for Cancers Resistant to Carbon Ion Radiotherapy"

_ijms, 2019, doi:10.3390/ijms20184563_

Round 1

Reviewer 1 Report

I would like to make just one minor comment. The results shown on Fig. 3C have not been explained in the text. It is not clear whether the three cell lines were exposed to ionising radiation before LY2874455 treatment or the graph shows the effects of this drug on non-irradiated cells. This should be clarified.

Reviewer 2 Report

This is an interesting study of carbon ion radiotherapy and suggesting that FGFR signaling is a target to understand resistance to carbon ion radiotherapy.

The central issue in the paper is why there is resistance. The increased RBE of carbon ion should result in a perfect situation for appropriate treatment planning, and prevent local recurrence. The issue with carbon ion radiotherapy as is the case with proton radiotherapy is the normal tissue tolerance. The authors should focus on normal tissue effects of carbon ion, rather than on cancer recurrence, which could be obviated by giving higher doses of irradiation.

The methodology is related to looking for genes possibly related to local recurrence. The studies should end with cell lines in culture and on Figure 4, the authors study A549, H1299, and H1703, which are cell lines. Studies with fresh tissue from patient biopsy specimens comparing pre-treatment and in-field recurrence would be appropriate for this study. In the absence of such data, the studies are theoretical and should be referred to as a potential therapeutic mechanism.

Round 2

Reviewer 2 Report

Accept as is.